# Biocontrol Potential of *Bacillus subtilis* and *Bacillus tequilensis* against Four *Fusarium* Species

**DOI:** 10.3390/pathogens12020254

**Published:** 2023-02-05

**Authors:** Vejonepher Baard, Olalekan Olanrewaju Bakare, Augustine Innalegwu Daniel, Mbukeni Nkomo, Arun Gokul, Marshall Keyster, Ashwil Klein

**Affiliations:** 1Plant Omics Laboratory, Department of Biotechnology, University of the Western Cape, Robert Sobukwe Road, Bellville 7530, South Africa; 2Department of Biochemistry, Faculty of Basic Medical Sciences, Olabisi Onabanjo University, Sagamu 121001, Nigeria; 3Environmental Biotechnology Laboratory, Department of Biotechnology, University of the Western Cape, Robert Sobukwe Road, Bellville 7530, South Africa; 4Department of Biochemistry, Federal University of Technology, P.M.B 65, Minna 920101, Nigeria; 5Department of Agriculture, University of Zululand, Main Road, KwaDlagezwe 3886, South Africa; 6Department of Plant Sciences, Qwaqwa Campus, University of the Free State, Phuthadithjaba 9866, South Africa

**Keywords:** antifungal, bacterial endophytes, biocontrol, phylogenetics, polymerase chain reaction

## Abstract

The use of biological control agents as opposed to synthetic agrochemicals to control plant pathogens has gained momentum, considering their numerous advantages. The aim of this study is to investigate the biocontrol potential of plant bacterial isolates against *Fusarium oxysporum*, *Fusarium proliferatum*, *Fusarium culmorum,* and *Fusarium verticillioides*. Isolation, identification, characterization, and in vitro biocontrol antagonistic assays of these isolates against *Fusarium* species were carried out following standard protocols. The bacterial endophytes were isolated from *Glycine max.* L leaves (B1), *Brassica napus.* L seeds (B2), *Vigna unguiculata* seeds (B3), and *Glycine max*. L seeds (B4). The bacterial isolates were identified using 16S rRNA PCR sequencing. A phylogenetic analysis shows that the bacterial isolates are closely related to *Bacillus subtilis* (B1) and *Bacillus tequilensis* (B2–B4), with an identity score above 98%. All the bacterial isolates produced a significant amount (*p* < 0.05) of indole acetic acid (IAA), siderophores, and protease activity. In vitro antagonistic assays of these isolates show a significant (*p* < 0.05) growth inhibition of the fungal mycelia in the following order: *F. proliferatum > F. culmorum > F. verticillioides > F. oxysporum,* compared to the control. The results suggest that these bacterial isolates are good biocontrol candidates against the selected *Fusarium* species.

## 1. Introduction

As the world population multiplies, there is a need to improve agricultural productivity and reduce global food production losses by controlling crop pests and disease outbreaks [1]. One of the major causes of significant losses in food production is crop pathogens [2,3]. Fungal pathogens are a substantial cause of crop losses, producing millions of spores that affect pre- and post-harvest crops [4]. The genus *Fusarium* is one of the most significant biological stressors affecting economically essential crops in South Africa [5]. The genus contains filamentous fungi from the phylum Ascomycota, and is found in soil, water, and air. They are one of the most diverse and widespread fungal pathogens, causing various diseases such as rots, blights, wilts, and cankers [6]. *Fusarium* species are known to infect a variety of staple crops such as corn, wheat, and soybeans worldwide, impacting the environment, animals, and human health. 

The major *Fusarium* species that infect many foods and feed crops include *F. oxysporum*, *F. culmorum, F. verticillioides,* and *F. proliferatum. F. oxysporum* is one of the most critical soil-borne plant pathogens as it infects many crops and is responsible for massive crop losses [7]. *F. culmorum* is known to infest wheat worldwide, causing wheat blight and crown rot of cereals, resulting in detrimental losses of this economically important crop [8,9]. *F. verticillioides* causes ear rot, stalk rot, and seedling blight in maize, resulting in significant economic losses in maize production worldwide [10]. *F. proliferatum* is associated with maize and soybean infections and has been reported to be a significant source of mycotoxin contamination in food [11]. The *Fusarium* species are known to produce mycotoxins such as deoxynivalenol, fumonisin-B1, moniliformin, and beauvericin, which are detrimental to plant, human, and animal health [12].

A key issue in agriculture is the sustainable reduction of the post-harvest yield across the food supply chain from harvest to consumption [13]. Several management strategies, including the use of agrochemicals, have been employed to minimize yield loss but have proved ineffective, due to the development of resistance as a result of continued usage, and the negative impact on the environment [14]. Copper-based fungicides can affect soil fertility and organisms due to the accumulation of copper in the soil, as copper is not biodegraded [15]. These chemicals used as fungicides are carcinogenic and can affect non-target organisms such as humans, animals, and other soil microbes such as nematodes, non-pathogenic bacteria, and fungi [16]. The role of these soil microbes includes the decomposition of organic matter, soil aeration, and the promotion of nutrient and water cycling in the soil, which has a positive effect on the soil composition and productivity [15]. 

Endophytes have long attracted interest as biocontrol agents (BCAs) due to their environmental friendliness, low cost, and reduced dependence on agrochemicals. Bacterial endophytes are efficient BCAs because they have adapted to live in plants and are self-sustaining [17]. Bacterial endophytes colonize the internal tissues of plants and live symbiotically without causing disease [18]. Endophytes benefit the host by stimulating plant growth and stress tolerance and protecting the host from disease and infection by pathogens [3]. They achieve this through mechanisms such as the competition for nutrients and space, the production of secondary metabolites, and plant hormone synthesis. These exudates or secondary metabolites play an active role in root–root and root–microbe interaction by influencing the biological and physical interactions between these interacting organisms. These metabolites can alter the physical, chemical, and biological properties of the soil and inhibit the growth of competing plant species [19]. Bacterial endophytes may use a combination of one or more of these metabolites to enhance pathogen inhibition and decrease the development of resistance [20].

The use of BCAs to control different *Fusarium* species has been reported [20]. Despite numerous studies on the inhibition potential of BCAs against the growth of the *Fusarium* species, there is a need to screen more biocontrol agents to add to the existing body of knowledge and increase awareness on their usage [21,22]. Therefore, this study aims to investigate the biological control potential of plant-associated bacterial endophytes against different *Fusarium* species in vitro. Several pathogenic soil-borne *Fusarium* species are susceptible to competition from plant-associated microbes [23]. The positive test for siderophore activity, protease activity, and chitinase activity could explain the mechanisms involved in suppressing the mycelial growth in the *Fusarium* species in this study.

## 2. Materials and Methods 

### 2.1. Isolation of Endophytic Bacteria

Bacterial endophytes were isolated from surface-sterilized canola (*Brassica napus* L.), cowpea (*Vigna unguiculata*) seeds, soybean (*Glycine max*) seeds, and leaves that were cultured on potato glucose agar (PGA) media. A single colony from each bacterial isolate was sub-cultured into Luria–Bertani (LB) broth and incubated overnight at 37 °C.

### 2.2. DNA Extraction and PCR Amplification of the 16S rRNA Gene

High molecular weight genomic DNA was extracted from the bacteria isolates using the acetyl trimethylammonium bromide (CTAB) procedure, previously described by Murray and Thompson [24], with slight modifications. A single colony from each bacterial isolate inoculated in LB broth was centrifuged at 13,000 rpm for 5 min using an Eppendorf microcentrifuge (5425 R, Stevenage, UK), and the pellet was re-suspended in 300 µL of 1 × Tris-EDTA (TE) buffer. The sample was incubated at 65 °C for 15 min with 50 µL of 10% sodium dodecyl sulfate (SDS). After incubation, 500 µL of 1.5 × CTAB buffer and 5 µL of 2-mercaptoethanol was added. The samples were incubated at 65 °C for 1 h before adding an equal volume of chloroform: isopropanol (24:1), and they were centrifuged at 13,000 rpm for 15 min using an Eppendorf microcentrifuge (5425 R, Stevenage, UK) before transferring the upper layer into a 2 mL Eppendorf tube. The DNA was precipitated with 0.6 mL ice-cold absolute ethanol for 1 h and centrifuged at 13,000 rpm for 2 min. The pellet was washed twice with 70% ice-cold ethanol and centrifuged at 13,000 rpm for 2 min. The pellet was allowed to air dry and was re-suspended in 10 µL of 1 × TE buffer until further use. The 16S rRNA gene was amplified by polymerase chain reaction (PCR) with universal primers: E9F (5′-GAG TTT GAT CCT GGC TCA G-3′) and U1510R (5′-GGT TAC CTT GTT ACA CTT-3′). A 25 µL PCR reaction mixture consisting of 2 µM forward primer, 2 µM reverse primer, 2 × Multiplex PCR Master mix (Qiagen, Hilden, Germany), and 25 ng/µL of genomic DNA was prepared. The control was prepared by adding distilled water instead of template DNA to the reaction mix. Samples were amplified using a T100^TM^ thermal cycler (BioRad, Hercules, USA) with initial denaturation at 95 °C for 15 min, 34 cycles at 94 °C for 30 s, 55 °C for 30 s, 72 °C for 30 s, the final extension at 72 °C for 5 min, and holding for 4 °C. PCR products were separated on a 1% agarose gel at 90 V.

### 2.3. DNA Sequencing and Data Analysis

The PCR amplicons of the 16S rRNA gene were sequenced using the BrilliantDye™ Terminator v3.1 Cycle Sequencing on an ABI3500xL genetic analyzer (Inqaba Biotechnical Industries (Pty) Ltd., Pretoria, South Africa). The obtained sequence data were edited and aligned using the Clustal W multiple sequence alignment tool in BioEdit sequence alignment editor version 7.05 (Raleigh, NC, USA) (http://www.mbio.ncsu.edu/BioEdit/bioedit.html (accessed on 15 September 2022)) to generate a contiguous consensus sequence. An aligned contiguous consensus sequence of 16S rRNA gene was used for homology search by the basic local alignment search tool (BLAST) software (http://blast.ncbi.nlm.nih.gov, accessed on 15 September 2022) algorithm at National Center for Biotechnology Information (NCBI) to identify bacterial isolates (to species’ level) based on highest percentage similarity. The phylogenetic analysis of the 16S rRNA gene sequences of the isolates (B1-4) including outgroup sequence (*Escherichia coli;* JCM 16946) was performed using the Molecular Evolutionary Genetics Analysis (MEGA) version X software [25]. Bootstrap analysis was performed in MEGA X using 100 replicates.

### 2.4. Gram Stain and Bacterial Morphology

A smear loop of each bacteria culture was heat fixed onto a glass slide and Gram stained [26]. The slide was viewed under a light microscope. Gram-positive bacteria are identified with a purple color, and Gram-negative bacteria are identified with a red/pink color. The morphology of the bacterium was also observed.

### 2.5. Catalase Test

Catalase activity of bacterial endophytes was tested by placing a smear loop of each culture onto a glass slide and adding 3% hydrogen peroxide to each isolate [27]. The formation of bubbles indicates a positive catalase activity.

### 2.6. Indole Acetic Acid (IAA) Production Test

Indole-3-acetic acid (IAA) production was determined using a colorimetric assay previously described by Gordon and Weber [28]. The endophytic bacterial isolates were grown in Yeast Extract Mannitol (YEM) broth (mannitol 1%, yeast extract 0.1%, dipotassium phosphate 0.05%, magnesium sulphate 0.02%, sodium chloride 0.01%) that was supplemented with 0.1% tryptophan and YEM broth without the addition of tryptophan. The cultures were grown at 37 °C with shaking for 5 days. *E. coli* Krx (Promega, Madison, WI, USA) was used as a control to screen for IAA production. After incubation, the cultures were centrifuged for 15 min at 13,000 rpm, and the supernatant was used for IAA extraction. The supernatant was mixed with Salkowski reagent (1:2) and incubated in the dark for 30 min at room temperature. The samples were measured at 530 nm using a FLUOstar Omega UV-visible spectrophotometer (BMG LabTech GmbH, Ortenberg, Germany) to determine the concentration of IAA. The absorbances were compared to a standard of IAA diluted in YEM broth with a concentration ranging from 5 µg/mL to 100 µg/mL.

### 2.7. Phosphate Solubilization Test

A smear loop of each bacterial culture was spot inoculated on Pikovskaya’s phosphate agar and grown for 7 days at room temperature [29]. A phosphate-solubilizing microorganism, *E. coli* Krx, was used as a positive control. After incubation, colonies were observed for halos, which indicates phosphate solubilization capability.

### 2.8. Siderophore Plate Assay

Siderophore activity was tested as described by Alexander and Zuberer [30]. A smear loop of each bacterial culture was spot inoculated on chrome *azurol* S (CAS) media and incubated at room temperature for 7 days. A bacterium with siderophore activity, *E. coli* Krx, was used as a positive control with *E. coli* XL gold (Agilent, Santa Clara, CA, USA) as a negative control. A clear zone around the colonies indicates siderophore activity.

### 2.9. Protease Activity Test

Protease production was determined according to the method described by Etminani and Harighi [31]. A smear loop of each isolate culture was spot inoculated on skim milk agar and incubated at room temperature for 4 days. *Bacillus licheniformis* (ATCC 145080)*,* a bacterium with protease activity, was used as a positive control while 7-day-old *E. coli* Krx was used as a negative control. Agar plates were monitored for the formation of a halo around the colonies.

### 2.10. Chitinase Activity Test

Chitinase activity was tested as described by Faramarzi et al. [32]. A smear loop of each bacterial isolate was spot inoculated onto the colloidal chitin agar and incubated at room temperature for 7 days. After incubation, agar plates were stained with 0.1% Congo red to visualize the positive clearance zone around the colonies.

### 2.11. Antifungal Activity of Bacterial Isolates 

The fungal pathogens used in this study were obtained from the fungal culture collection from the Plant Protection Institute, Agricultural Research Council (ARC), Pretoria, South Africa. All *Fusarium* species were grown and maintained on potato glucose agar (PGA) at room temperature (25 ± 2 °C). A well-diffusion method was used to screen bacterial isolates for their ability to suppress the growth of four *Fusarium* species, *F. oxysporum* PPRI 19027, *F. proliferatum* MRC 2059, *F. culmorum* PPRI 10138, and *F. verticillioides* MRC 826. For each fungal strain, an agar plug (1.0 cm × 1.0 cm) of 7-day-old actively growing mycelia was placed at the center of a new PGA plate and was allowed to grow at 30 °C for 48 h. After 48 h, 100 µL of each bacterial endophyte culture was inoculated on PGA plates at four equidistant sites (5 mm wells), 3 cm apart from the *Fusarium* colonies in the center. The control plates were inoculated with 0.03% carbendazim, a commercial fungicide. All plates were incubated at 30 °C for approximately 12 days. The antagonistic effect of bacterial isolates on *Fusarium* species’ mycelial growth was measured on day 12.

### 2.12. Statistical Analysis

The data collected were analyzed using one-way analysis of variance (ANOVA) while treatment means were separated by the least significance difference (LSD) incorporated in the Statistical Package for Social Sciences (SPSS) (version 26, IBM, Armonk, NY, USA).

## 3. Results

### 3.1. PCR Amplification of the Bacterial Isolates

The extracted high molecular weight genomic DNA was used for identifying all bacterial isolates via PCR amplification and sequencing of the 16S rRNA gene. Figure 1 shows the positive PCR amplification of two *Bacillus* controls (*Bacillus subtilis* ATCC^®^ 19659™ and *Bacillus tequilensis* NCTC 13306) and the bacterial isolates (B1, B2, B3, and B4) used in this study. The samples produced a band at 1500 bp when amplified using the 16S rRNA gene marker, with no visible signs of degradation or the presence of primer dimers. 

### 3.2. Molecular Identification and Phylogenetic Analysis of Bacterial Isolates

The bacterial endophytes isolated from soybean leaves, canola, cowpea, and soybean seeds showed a high sequence similarity to *Bacillus* species, using the basic local alignment search tool (BLAST) against the non-redundant NCBI database (Table 1). A molecular analysis of isolates B1 and B2–B4 revealed a close phylogenetic relationship to *Bacillus subtilis* and *Bacillus tequilensis*, respectively (Figure 2). Based on the BLAST search against NCBI, isolate B1 shows a 98.35% similarity with the *Bacillus subtilis* strain S-8. In addition, isolate B2 shows similarity to the *Bacillus tequilensis* strain 127 (98.69%), whereas B3 and B4 showed similarity with *Bacillus tequilensis* km24 (97.84%) and *Bacillus tequilensis* A37 (98.50%), respectively (Table 1). The maximum likelihood approach was used to determine the phylogenetic relationship between the identified isolates and selected database sequences (Figure 2).

### 3.3. Biochemical Characterization of Bacterial Isolates

Four bacterial isolates (B1–B4) including a positive control, *E. coli* Krx, were quantitatively screened to produce IAA using tryptophan as a precursor for IAA. The result of the IAA production potentials of the bacterial isolates revealed that B3 produced the highest amount of IAA (5.05 μg/mL), followed by isolate B4 (5.02 μg/mL), which are significantly higher (*p* < 0.05) than the control strain (*E. coli* Krx), which produced 3.21 µg/mL IAA (Figure 3).

Catalase activity of all the isolates shows that they are all positive to the test while the Gram stain reaction shows that the isolates are all Gram-positive with rod shapes (Table 2). The phosphate solubilization results showed that none of the bacterial isolates showed zones of clearance and, as a result, could not solubilize phosphate, relative to the positive control (*E. coli* Krx), which produced a clear halo around the bacteria (Table 2; Figure 4).

The isolates were capable of siderophore production, as indicated by clearance zones around the B1–B4 colonies (Figure 5). However, all isolates showed siderophore activity with a narrow zone of clearance ranging between 16.50 ± 0.25 and 21.20 ± 1.27 mm, which were significantly comparable (*p* < 0.05) with the control (21.20 ± 0.10 mm), as shown in Table 2. 

All bacterial isolates, except B1, could produce protease activity, as indicated by the clearance zones around the colonies (Figure 6). A similar zone of clearance was produced by all the isolates, with B3 having the least zone of clearance (11.60 ± 0.42 mm), relative to the control (15.30 ± 0.75 mm) (Table 2).

The chitinase test shows that isolates B2 (5.10 ± 0.42 mm) and B3 (6.13 ± 0.71 mm) had a chitinase-producing ability with a clear zone around the colonies (Figure 7). This shows that these bacterial isolates have the potential to degrade chitin, which forms an integral part of fungal cell walls. However, no chitinase activity was detected for isolates B1 and B4. 

### 3.4. Biocontrol Activity of the Bacterial Isolates against Fusarium Species

Biocontrol activity of the bacterial isolates against four different *Fusarium* species shows significant inhibition (*p* < 0.05) of the fungi (Table 3, Figure 8) with *F. proliferatum* as the most inhibited by isolates B3 and B4 with a percentage inhibition of 60.75 ± 0.21 and 64.79 ± 0.40%, respectively. The results showed that all bacterial isolates significantly (*p* < 0.05) inhibit the mycelial growth of all the four *Fusarium* species (*F. culmorum*, *F. proliferatum* and *F. verticillioides*), preventing them from growing more than 3 cm in diameter, which was comparable to the positive control (carbendazim). 

## 4. Discussion

*Bacillus* species have been well-documented for their ability to enhance plant growth and protect host plants against pathogen infections [19]. Bacterial endophytes were isolated from some plant seeds and leaves to evaluate their biocontrol potential. The bacterial endophytes isolated from soybean leaves, canola, cowpea, and soybean seeds showed a high sequence similarity to *Bacillus* spp, using the BLAST analysis from the NCBI database. Isolates B1 and B2-B4 revealed a close phylogenetic relationship to *Bacillus subtilis* and *Bacillus tequilensis*, respectively (Table 1; Figure 2).

The bacterial endophytes possessed IAA activity using tryptophan as a precursor. IAA is known to aid cell division, elongation, and the overall growth of plants [33]. IAA may also indirectly improve phosphorus acquisition by increasing plant root growth [34]. All the bacterial isolates significantly produced IAA (*p* < 0.05), which is comparable with the control (Figure 3). According to a report by Patten and Glick [35], 1 nM to 1 pM of IAA could promote plant growth, which suggests that the amount of IAA produced by these bacterial isolates in this study is sufficient to influence plant growth. 

Phosphorus is a macronutrient required for the growth and development of plants. Bacteria capable of solubilizing insoluble phosphate can increase soil quality and plant growth [31]. The result of this study shows that there is no evidence of phosphate solubilization by any of the bacterial isolates (Figure 4). The inability of the bacterial isolates to solubilize phosphate implies that they could not utilize this mechanism of phosphate solubilization pathways to inhibit the growth of the *Fusarium* species. The result presented in this study is in agreement with the findings of Almoneafy et al. [36], who reported the inability of *B. subtilis* strain D16 to solubilize phosphate.

Siderophore production is essential for adapting the plant to an iron-limiting environment and controlling phytopathogens by inhibiting iron availability to pathogens [37]. The result of this study shows that all the bacterial isolates produced significant siderophore activity, which is significantly lower (*p* < 0.05) than the control organism (Table 2; Figure 5). Siderophore production for iron competition has been recognized as a significant antagonistic trait in many BCAs against phytopathogens [38]. Some fungal pathogens may also synthesize siderophores; however, they have a much lower affinity than bacteria. As a result, these bacterial isolates can potentially outcompete pathogens such as the *Fusarium* species for the available iron [39]. 

Lytic enzymes, such as chitinases and proteases, can degrade fungal cell walls and inhibit fungal growth in vitro, showing direct antifungal properties [38]. In this study, except for B1, all the bacterial isolates have protease activity (Table 2; Figure 6), while only B2 and B3 show chitinase activity (Table 2; Figure 7). Several bacterial genera, such as *Bacillus*, *Pseudomonas,* and *Serratia* are known to suppress fungal pathogens by producing lytic enzymes [39]. Proteases are enzymes that hydrolyze peptide bonds between amino acids [40]. Most fungal cell walls contain chitin [41], which is broken down by bacterial lytic enzymes, such as chitinases, to utilize the dead material as nutrients [42]. Ramyabharathi and Raguchander [38] reported that the bacterial endophyte *B. subtilis* EPCO16 inhibits the mycelial growth of *F. oxysporum* in vitro by producing chitinase and protease. The isolate B1 from the current study did not produce protease activity because it could not hydrolyze peptide bonds between amino acids of the fungi using protease pathways; isolates B1 and B4 did not produce chitinase and would not be able to inhibit the growth of the *Fusarium* species using disruption of the fungal chitin walls but still inhibited the *Fusarium* species’ growth. This suggests that B1 and B4 can utilize other molecules or pathways to suppress the growth of fungal pathogens, such as the production of volatile compounds [43].

The mycelial growth of the four *Fusarium* strains, *F. oxysporum* PPRI 19027, *F. culmorum* PPRI 10138, *F. proliferatum* MRC 2059, and *F. verticillioides* MRC 826, was significantly inhibited (*p* < 0.05) by all the bacterial isolates after 12 days (Table 3, Figure 8). The inhibitory capacity of the bacterial isolates could be attributed to their ability to grow faster and suppress the fungal pathogens by efficiently competing for space and nutrients. Soil-borne pathogens, such as *Fusarium,* that infect plants through mycelia contact are generally more susceptible to competition by plant-associated microorganisms [23]. According to Bacon et al. [44], *F. verticillioides* infection in maize plants could be inhibited by *Bacillus subtilis* because they occupy the same ecological niche based on competitive exclusion. 

## 5. Conclusions

In this study, we report the biocontrol potential of four plant bacterial isolates against *F. oxysporum, F. proliferatum, F. culmorum,* and *F. verticillioides.*

The bacterial isolates were identified as *Bacillus subtilis* and *Bacillus tequilensis,* which displayed various indirect mechanisms for plant growth promotion, including IAA production, extracellular hydrolytic enzyme activities, and in vitro antagonistic activities against four phytopathogenic *Fusarium* strains. Endophytic bacteria with plant-growth-promoting characteristics may provide a solution to sustainably improve crop yield. The results further showed that the isolated bacterial endophytes significantly inhibited the growth of the *Fusarium* species in this order: *F. proliferatum > F. culmorum > F. verticillioides > F. oxysporum* in the in vitro biocontrol assay. From this study, we conclude that the inhibitory activity of the bacterial isolates may be attributed to their ability to produce siderophore, protease, catalase, and chitinase activities, which have been reported to cause the inhibition of fungal pathogens. Although these attributes may contribute to the antifungal activity exhibited by the bacterial isolates, more research is needed to fully characterize the antimicrobial capacity of these isolates by evaluating their ability to produce volatile microbial compounds and other lytic enzymes, such as β-1,3-glucanase.

## Figures and Tables

**Figure 1 pathogens-12-00254-f001:**
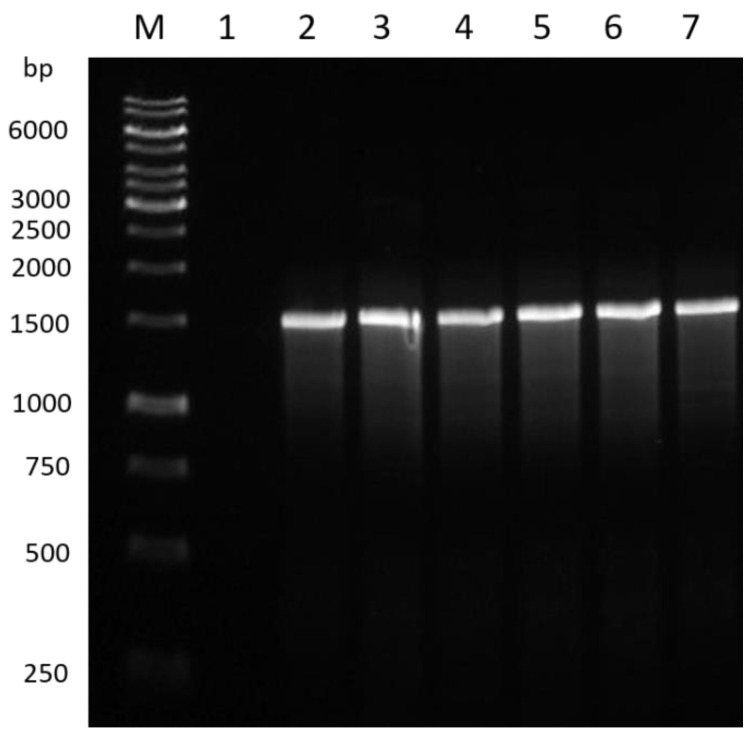
PCR amplification products of the 16S rRNA gene marker for *Bacillus* positive controls and bacterial isolates (B1–B4) used in this study. (Lane M: GeneRuler 1 kb DNA ladder; lane 1: negative control; whereas lane 3: *Bacillus subtilis* ATCC^®^ 19659™; lane 4: *Bacillus tequilensis* NCTC 13306; lane 5: B1; lane 6: B2; lane 7: B3; and lane 8: B4).

**Figure 2 pathogens-12-00254-f002:**
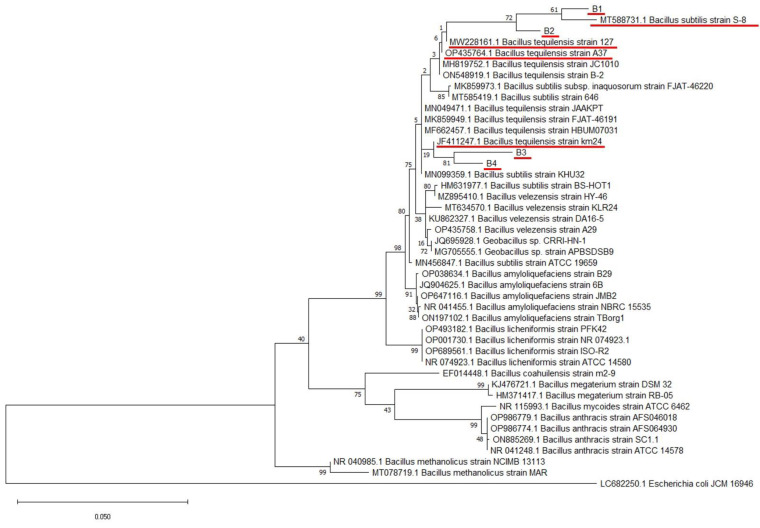
Phylogenetic analysis of 16s rRNA sequences of isolates B1, B2, B3, and B4, with reference sequences from NCBI (Table 1) using the MEGA maximum likelihood method based on the Tamura–Nei model. The bootstrap values are expressed as a percentage of 100 replicates. *Escherichia coli* JCM 16946 was used as outgroup. The identified bacterial isolates and their top hits are underlined in red.

**Figure 3 pathogens-12-00254-f003:**
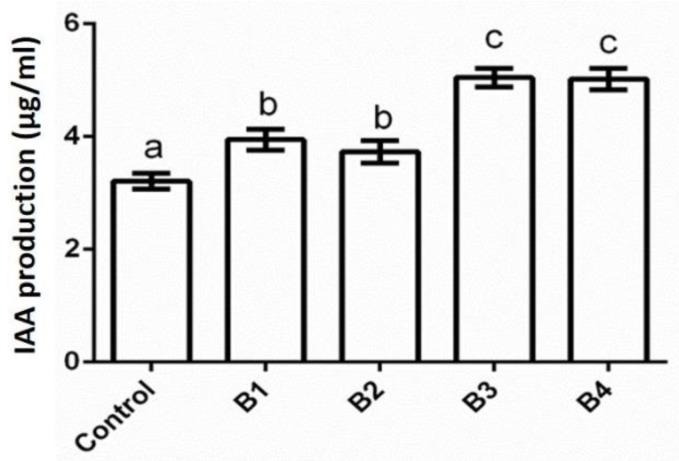
IAA production of bacterial isolates. The letters a, b, c indicate statistically significant difference at *p* < 0.05. Bars with different letters are significantly different at *p* < 0.05.

**Figure 4 pathogens-12-00254-f004:**
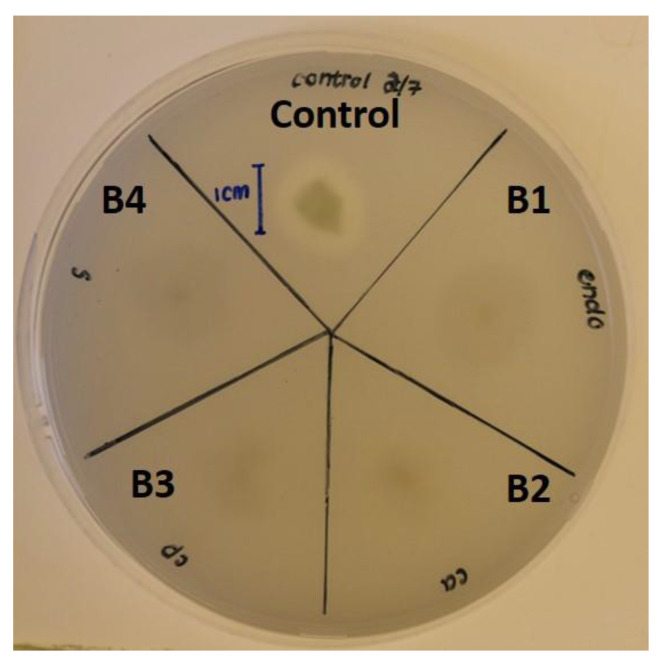
Phosphate solubilization test showing isolates B1–B4 and control spot inoculated onto Pikovskayas agar. B1 represents *Bacillus subtilis*, whereas B2, B3, and B4 represent *Bacillus tequilensis*.

**Figure 5 pathogens-12-00254-f005:**
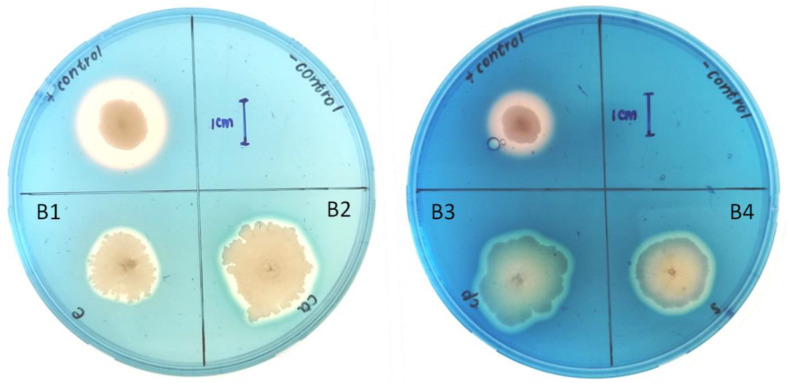
Siderophores activity of bacterial isolates on chrome azurol S media plates. B1 represents *Bacillus subtilis*, whereas B2, B3, and B4 represent *Bacillus tequilensis*.

**Figure 6 pathogens-12-00254-f006:**
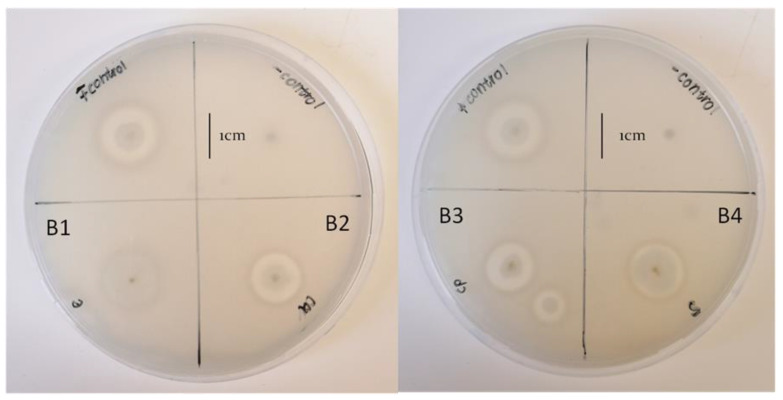
Protease-producing ability of bacterial isolates B1-B4 on skim milk agar as indicated by a zone of clearance around their colonies. B1 represents *Bacillus subtilis*, whereas B2, B3, and B4 represent *Bacillus tequilensis*.

**Figure 7 pathogens-12-00254-f007:**
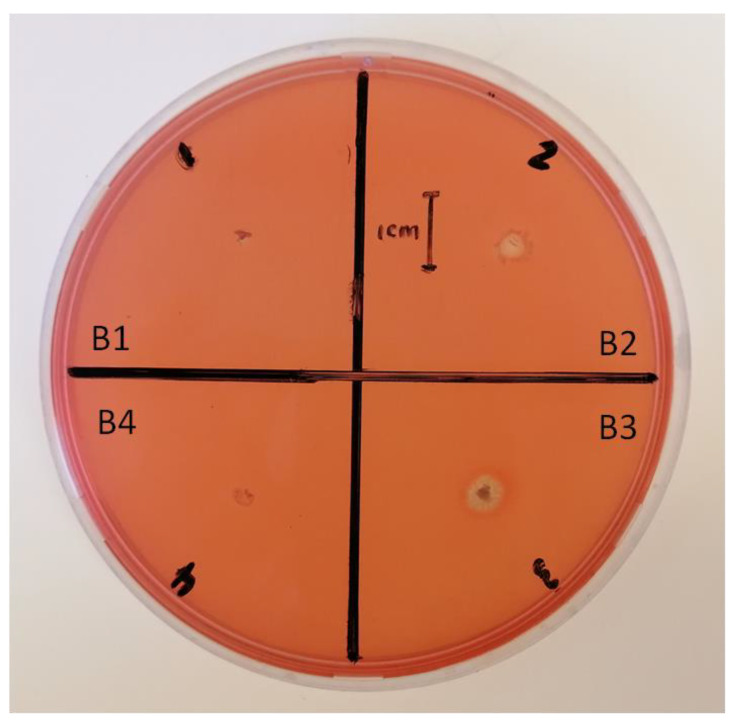
Chitinase-producing ability of the bacterial isolates grown on chitin agar. B1 represents *Bacillus subtilis*, whereas B2, B3, and B4 represent *Bacillus tequilensis*.

**Figure 8 pathogens-12-00254-f008:**
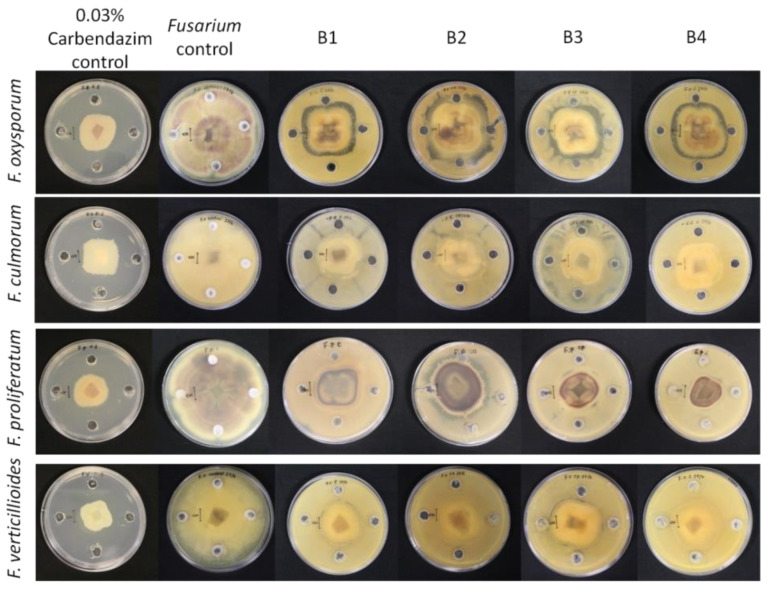
Biocontrol potential of bacterial isolates against four *Fusarium* species. B1 represents *Bacillus subtilis*, whereas B2, B3, and B4 represent *Bacillus tequilensis*.

**Table 1 pathogens-12-00254-t001:** Identification of bacterial isolates based on 16S rRNA gene sequencing.

Bacterial Isolates	Host	Tissue	Top BLAST Hit	Accession Number	Sequence Identity (%)
B1	*Glycine max* L.	leaves	*Bacillus subtilis* strain S-8	MT588731.1	98.35
B2	*Brassica napus* L.	seeds	*Bacillus tequilensis* strain 127	MW228161.1	98.69
B3	*Vigna unguiculata*	seeds	*Bacillus tequilensis* strain km24	JF411247.1	97.84
B4	*Glycine max* L.	seeds	*Bacillus tequilensis* strain A37	OP435764.1	98.50

**Table 2 pathogens-12-00254-t002:** Biochemical characterization of the endophytic bacteria isolated from different plant tissue.

Bacterial Isolates	Catalase Activity	Gram Staining	Shape/Type	Phosphate Solubilization	Siderophores Activity	Protease Activity	Chitinase Activity
Control	+	+	Rod	20.22 ± 0.51	21.20 ± 0.10 ^b^	15.30 ± 0.75 ^b^	−
B1	+	+	Rod	−	16.50 ± 0.25 ^a^	15.50 ± 0.43 ^b^	−
B2	+	+	Rod	−	21.20 ± 1.27 ^b^	12.30 ± 0.81 ^a^	5.10 ± 0.42 ^a^
B3	+	+	Rod	−	20.90 ± 0.93 ^b^	11.60 ± 0.42 ^a^	6.13 ± 0.71 ^a^
B4	+	+	Rod	−	18.00 ± 0.97 ^ab^	12.80 ± 0.30 ^a^	−

The letters ^a, b^ indicate statistically significant difference at *p* ≤ 0.05. Values (clearance zones in mm) within the same column with different letters (^ab^) are significantly different (*p* ≤ 0.05), values are the means ± SE (n = 3). + denotes positive test and − denotes negative test.

**Table 3 pathogens-12-00254-t003:** Percentage inhibition of *Fusarium* mycelia by the bacterial isolates.

Fungal Species	B1	B2	B3	B4	Carbendazim
*F. oxysporum*	40.64 ± 0.38 ^b^	35.54 ± 0.18 ^a^	44.36 ± 0.48 ^b^	39.48 ± 0.46 ^ab^	59.66 ± 0.40 ^c^
*F. culmorum*	54.06 ± 0.05 ^ab^	55.65 ± 0.38 ^ab^	51.94 ± 0.52 ^a^	50.49 ± 0.16 ^a^	56.45 ± 0.73 ^b^
*F. proliferatum*	58.74 ± 0.09 ^b^	45.67 ± 0.99 ^b^	60.75 ± 0.21 ^b^	64.79 ± 0.40 ^b^	60.89 ± 0.34 ^b^
*F. verticillioides*	57.49 ± 0.23 ^c^	51.73 ± 0.26 ^b^	43.29 ± 0.23 ^a^	53.39 ± 0.02 ^c^	64.68 ± 0.13 ^d^

The letters ^a, b, c, d^ indicate statistically significant difference at *p* ≤ 0.05. Values (clearance zones in mm) within the same column with different letters (^ab^) are significantly different (*p* ≤ 0.05), values are the means ± SE (n = 3). + denotes positive test and − denotes negative test.

## Data Availability

Not applicable.

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
