# Peer review of "Biocontrol Potential of Bacillus subtilis and Bacillus tequilensis against Four Fusarium Species"

_pathogens, 2023, doi:10.3390/pathogens12020254_

Round 1

Reviewer 1 Report

The article by Baard et al., titled: Biocontrol potential of Bacillus subtilis and Bacillus tequilensis against four Fusarium species is well written. The study evaluated 4 biocontrol strains, one B. subtilis, and three strains belonging to B. tequilensis for their in vitro inhibitory effect against mycelial growth of 4 Fusarium species which showed susceptibility in the order; F. proliferatum > F. culmorum > F. verticilloides > F. oxysporum. I suggest minor revisions before it can be published.

The following suggestions are for general minor revisions. Specific suggestions are provided in the pdf file.

1.      The authors hypothesize that the antifungal activity of the bacterial strains is due to the siderophores activity, protease activity, and chitinase activity as demonstrated by the qualitative in vitro results. Even though the methods and the results of these three parameters are quite elaborate, the authors could have left out various potential attributes of these biocontrol isolates, including the production of other lytic enzymes like β-1,3-glucanase, the potential production of antibiotics, etc. Therefore, it is important to be cautious in your conclusions, to indicate that these three attributes could have contributed to the observed antifungal activity of these biocontrol strains and highlight the gaps for further evaluation of the antifungal attributes of these strains.

2.      The author used a single bacterial colony (instead of bacterial culture broth (stock) from a single colony) for each strain to assess each of the three parameters. But there is a possibility of dissociation in members of bacillus genera and potential variability when different colonies are used during the study. How did the authors ensure uniformity?

3.      And the authors mention the zone of clearance being significantly lower (p < 0.05) than the control in figures 5 – 7, please state the values of the measured clear zone.

4.      In Figure 5, please enhance the image of the first plate (B1 and B2) to make the clear zone more visible.  Same for Figure 7.

5.      In figure 1, lanes 3 and 5 are for B. subtilis standard and sample, respectively, while lanes 4 and 6 – 8 are for B. tequilensis standard and sample, respectively. However, it’s hard to notice the difference. If possible, provide a better picture or provide a brief explanation to guide the readers to interpret the picture.

6.      Figure 2, it is better to construct 4 separate phylogenetic trees for each isolate. B3 and B4 seem to be almost identical, including in their bioactivities, except for chitinase activity. Constructing separate trees could help to reveal the difference, in terms of the closest strains from the database.

7.      Table 2, did you replicate the plates for each group? If so, there is no need for taking duplicate readings, you can report average inhibitory values ± standard error of n number of replicates.

8.      Make sure to provide the results obtained in each of the described methods. For example, the results for the catalase test and gram staining have not been presented or mentioned in the results.

9.      In both your abstract and conclusion, highlight the aspect of plant growth-promoting potential of the isolated strains as demonstrated by IAA production

10.   Your study is good but it is not exhaustive. So, in your conclusion, please highlight other possible attributes of these isolates that could be responsible for the observed antifungal effect. Mention them as areas for further study. For example, the last paragraph of your discussion section can be briefly summarized and presented as an area for further study.

11.   Please double-check that all references cited in the text are relevant to the particular argument (before re-submitting your article).

Good luck

Author Response

Response to Reviewer 1 Comments

Point 1: The authors hypothesize that the antifungal activity of the bacterial strains is due to the siderophores activity, protease activity, and chitinase activity as demonstrated by the qualitative in vitro results. Even though the methods and the results of these three parameters are quite elaborate, the authors could have left out various potential attributes of these biocontrol isolates, including the production of other lytic enzymes like β-1,3-glucanase, the potential production of antibiotics, etc. Therefore, it is important to be cautious in your conclusions, to indicate that these three attributes could have contributed to the observed antifungal activity of these biocontrol strains and highlight the gaps for further evaluation of the antifungal attributes of these strains.

Response 1: The points raised by the reviewer is very important and have been critically addressed. The conclusion section of the manuscript was substantially revised to address the concerns raised by the reviewer.

Point 2: The author used a single bacterial colony (instead of bacterial culture broth (stock) from a single colony) for each strain to assess each of the three parameters. But there is a possibility of dissociation in members of bacillus genera and potential variability when different colonies are used during the study. How did the authors ensure uniformity?

Response 2: This observation was corrected accordingly. A loop full of the bacterial culture was used in each of the assay and not a single colony as was earlier reported.

Point 3: And the authors mention the zone of clearance being significantly lower (p < 0.05) than the control in figures 5 – 7, please state the values of the measured clear zone.

Response 3: The zones of inhibition have been measured and presented in table 2. The unit of the clear zones are converted to mm for clarity and convenience.

Point 4: In Figure 5, please enhance the image of the first plate (B1 and B2) to make the clear zone more visible.  Same for Figure 7.

Response 4: I think the reviewer meant to say figure 4 instead of figure 5. There was no phosphate solubilization activity for any of the bacteria isolates as stated in the table and so the zone of clearance were not present except for the positive control which is clearly visible.

Point 5: In figure 1, lanes 3 and 5 are for B. subtilis standard and sample, respectively, while lanes 4 and 6 – 8 are for B. tequilensis standard and sample, respectively. However, it’s hard to notice the difference. If possible, provide a better picture or provide a brief explanation to guide the readers to interpret the picture.

Response 5: The image in Figure 1 have been re labelled to be more descriptive for each of reference. All the necessary corrections have been effected in the revised manuscript.

Point 6: Figure 2, it is better to construct 4 separate phylogenetic trees for each isolate. B3 and B4 seem to be almost identical, including in their bioactivities, except for chitinase activity. Constructing separate trees could help to reveal the difference, in terms of the closest strains from the database.

Response 6: The authors would like to thank the reviewer for the suggestion. However, the homology search for each bacterial isolate did reveal the closest strain match for each isolate (documented in Table 1). 

Point 7: Table 2, did you replicate the plates for each group? If so, there is no need for taking duplicate readings, you can report average inhibitory values ± standard error of n number of replicates.

Response 7: The table have now been corrected to Table 3. The statistical analysis description for the results presented in the table have been revised.

Point 8: Make sure to provide the results obtained in each of the described methods. For example, the results for the catalase test and gram staining have not been presented or mentioned in the results.

Response 8: The result of the catalase activity and Gram stain test have been included in the revised manuscript as Table 2.

Point 9: In both your abstract and conclusion, highlight the aspect of plant growth-promoting potential of the isolated strains as demonstrated by IAA production

Response 9: It has been included in both sections as recommended.

Point 10: Your study is good but it is not exhaustive. So, in your conclusion, please highlight other possible attributes of these isolates that could be responsible for the observed antifungal effect. Mention them as areas for further study. For example, the last paragraph of your discussion section can be briefly summarized and presented as an area for further study.

Response 10: The conclusion section of the manuscript was substantially revised to address the concerns raised by the reviewer.

Point 11: Please double-check that all references cited in the text are relevant to the particular argument (before re-submitting your article).

Response 11: All the references used in manuscript was check against each documented argument and updated accordingly.

Reviewer 2 Report

This is an interesting manuscript about the biocontrol potential of some bacterial endophytes against four species of Fusarium. The manuscript shows new results, and the hypotheses are clear. This manuscript may be acceptable from my point of view as a limitedly focused paper. However, I have some points that need to be addressed in this manuscript.

My revision suggestion includes the following:

1. Line 36: Change to ''and reduce global food production''.

2. Line 66: Change to ''microfauna include''.

3. Line 90-91: All scientific names must be written in italics

4. Line 175: Please add the source and codes of Fusarium species or give detailed information about the isolation and identification of these strains.

5. Please add a part about the statistical analysis in the “Materials and Methods” section.

6. All information necessary to understand the table and figure should be included in the title or caption. For example, the information related to the statistical analysis is not completed in Table 2, and Figure 3.

7. Most of the obtained results need more deep discussion.

Author Response

Response to Reviewer 2 Comments

Point 1: Line 36: Change to ''and reduce global food production''.

Response 1: Corrected. A space was inserted between ‘and’ and ‘reduce’ in the sentence.

Point 2: Line 66: Change to ''microfauna include''.

Response 2: Corrected. A space was inserted between microfauna’ and ‘include’.

Point 3: Line 90-91: All scientific names must be written in italics

Response 3: The entire manuscript was checked and all scientific names have been italicised.

Point 4: Line 175: Please add the source and codes of Fusarium species or give detailed information about the isolation and identification of these strains.

Response 4: The origin (source) and assigned codes for each Fusarium species used in this study have been added to the revised manuscript.

Point 5: Please add a part about the statistical analysis in the “Materials and Methods”

Response 5: The statistical analysis for Table 2 (now Table 3) and Figure 3 have been completed in the revised manuscript.

Point 6: All information necessary to understand the table and figure should be included in the title or caption. For example, the information related to the statistical analysis is not completed in Table 2, and Figure 3.

Response 6: The statistical analysis for Table 2 (now Table 3) and Figure 3 have been completed in the revised manuscript.

Point 7: Most of the obtained results need more deep discussion.

Response 7: Thank you for the suggestion. We have revised the discussion section of the manuscript in line with all reviewer comments and have made the necessary changes.

Round 2

Reviewer 2 Report

Thanks for considering the suggestions from the previous version. The manuscript is acceptable from my point of view. I recommend accepting it for publication.

Author Response

Response to Academic Editor Notes

Point 1: Thanks for considering the suggestions from the previous version. The manuscript is acceptable from my point of view. I recommend accepting it for publication.

Response 1: We appreciate Reviewer 2 for the time taken to consider our manuscript for review and publication in the journal.